# A distinct transition from cell growth to physiological homeostasis in the tendon

Mor Grinstein[1], Heather L Dingwall[2], Luke D O'Connor[1], Ken Zou[1], Terence Dante Capellini[2,3], Jenna Lauren Galloway[1,4]*

[1]Center for Regenerative Medicine, Department of Orthopaedic Surgery, Massachusetts General Hospital, Harvard Medical School, Boston, United States; [2]Department of Human Evolutionary Biology, Harvard University, Cambridge, United States; [3]Broad Institute of Harvard and MIT, Cambridge, United States; [4]Harvard Stem Cell Institute, Cambridge, United States

**Abstract** Changes in cell proliferation define transitions from tissue growth to physiological homeostasis. In tendons, a highly organized extracellular matrix undergoes significant postnatal expansion to drive growth, but once formed, it appears to undergo little turnover. However, tendon cell activity during growth and homeostatic maintenance is less well defined. Using complementary methods of genetic H2B-GFP pulse-chase labeling and BrdU incorporation in mice, we show significant postnatal tendon cell proliferation, correlating with longitudinal Achilles tendon growth. Around day 21, there is a transition in cell turnover with a significant decline in proliferation. After this time, we find low amounts of homeostatic tendon cell proliferation from 3 to 20 months. These results demonstrate that tendons harbor significant postnatal mitotic activity, and limited, but detectable activity in adult and aged stages. It also points towards the possibility that the adult tendon harbors resident tendon progenitor populations, which would have important therapeutic implications.

DOI: https://doi.org/10.7554/eLife.48689.001

*For correspondence:
JGALLOWAY@mgh.harvard.edu

**Competing interests:** The authors declare that no competing interests exist.

## Introduction

Development, growth, and homeostasis rely on the precise regulation of cell proliferation and differentiation to generate and maintain a functioning organism. Frequent cell divisions grow tissues to the proper size, as do modifications to non-cellular tissue properties such as to the extent of extracellular matrix. However, once size is achieved, each tissue maintains its physiological functionality either through stem cell-mediated mechanisms as in the intestine (*Simons and Clevers, 2011*), the duplication of specialized cell types as in the liver (*Miyajima et al., 2014*), or in the virtual absence of cell division as in the heart (*Senyo et al., 2014*). In some cases, the transition from active proliferating to terminally differentiated cells has been attributed to a change in regenerative potential as observed in neonate verses adult mouse hearts (*Senyo et al., 2014*). Therefore, understanding transitions in cell turnover are important for setting the framework for more deeply understanding proliferative-driven growth stages and distinguishing between specific homeostatic renewal mechanisms in the adult. This knowledge is significant in considering therapies for tendon injuries, which can be challenging to treat due to their imperfect healing and propensity for re-injury (*Thomopoulos et al., 2015*).

Tendons begin as aggregations of cells that secrete and organize a highly ordered matrix to connect the musculoskeletal system and enable movement. Therefore, tendon growth and maintenance must not only involve its matrix but also the cells that generate and eventually reside within it, making knowledge of transitions in cell division important for understanding these processes. In the adult, the tendon matrix contains organized type I collagen fibrils and tenocytes, which are mature

**eLife digest** Muscles are anchored to bones via fibrous structures called tendons, which are made from specialized cells and large proteins called collagens. Tendons in newborns have round cells that can easily divide to make new cells, allowing the tissue to grow. Compared to newborns, tendon cells in adults have a star-like shape and appear to have stopped dividing. Tendon cells in adults are less abundant than in newborns, with more of the tendon made up of collagen. Adult tendons also do not heal as well as young tendons following an injury. However, it was previously unknown when the characteristics of young tendons are lost. Additionally, it was unclear whether adult tendon cells completely stop dividing or simply do so more slowly, or if these changes in cell division are what causes adult tendons to heal less easily.

To investigate this, Grinstein et al. used microscopy and cellular and molecular tools to examine tendon cell division in mice of different ages. The results indicated that tendon cells continue to divide after birth, but they do so more slowly as mice age. The researchers saw the most significant changes in the mice after they reached two weeks of age, which is also when the structure of the tendons begins to transition to that of an adult.

Young mice, which are growing and learning to move, have tendons that produce new cells faster than adult tendons. This may be necessary for young tendons to grow and adapt to the strains of movement. As tendons age, most of their cells lose the ability to divide, which seems to prevent the tendons from healing fully after injury. Notably, Grinstein et al. found that some cells are still able to divide slowly in adult tendons, which may be important for healing.

Further research is needed to examine the mechanisms involved in these changes and the factors that drive them, but a deeper understanding of tendon biology could lead to new therapies for treating tendon injuries.

DOI: https://doi.org/10.7554/eLife.48689.002

tendon cells possessing cellular extensions that project into the matrix (*Kalson et al., 2015*; *Kannus, 2000*). This mature stellate morphology differs greatly from that of the rounded shape of embryonic and neonatal tenoblasts. During embryogenesis, limb bud mesenchymal cells express the transcription factor, *Scleraxis* (*Scx*), and coalesce into tendon primordia, which organize to connect muscle and bone (*Schweitzer et al., 2001*). Through the transgenic labeling of cell cycle state using the Fluorescent ubiquitination-based cell cycle indicator (Fucci), robust numbers of mitotic tendon cells have been detected prior to birth (*Esteves de Lima et al., 2014*). Specific segments of the limb display more proliferative activity than others (*Huang et al., 2015*), suggesting that there are localized effects on tendon cell proliferation during embryogenesis. In addition to cell growth, these embryonic stages are marked by an increase in the number of collagen fibrils deposited in the matrix (*Kalson et al., 2015*). These collagen fibrils grow in length and diameter to grow the tissue (*Ezura et al., 2000*). Scanning electron microscopy at postnatal stages (P0 and P42) has shown an increase in the diameter of the collagen fibrils rather than an increase in collagen fibril or cell number in the tail tendons of mice (*Kalson et al., 2015*). These observations have led to a model whereby tendon postnatal growth is primarily driven by expansion of the extracellular matrix (ECM), which results in a reduction in cell density across the whole tissue in growth and aging (*Dunkman et al., 2013*; *Kalson et al., 2015*). However, a direct analysis of cell turnover and the transition in proliferative activity from birth to adult and aged stages has not been performed.

Although adult tendons display limited proliferation, some cell division has been detected in vitro and in vivo, especially in the context of injury. Tendon-derived stem/progenitor cells were characterized based on their ex vivo abilities to proliferate, clonally expand, and undergo serial transplantation (*Bi et al., 2007*). However, the identity and in vivo activity of the resident cell population remains unknown. Other studies have reported proliferation in adult tendons during homeostasis and repair (*Lindsay and Birch, 1964*; *Runesson et al., 2013*; *Tan et al., 2013*). Because these studies were performed without genetic lineage tracing tools, the origin of the cells proliferating in response to injury was unclear. Recent lineage tracing experiments have shown that in multiple cell-lineages, including *alpha-Smooth muscle actin* (*Acta2*), *Scx*, and *S100a4* lineage cells, can contribute to the healing tissue depending on the tendon and injury type, suggesting these populations may

retain proliferative abilities in adult tendons (*Best and Loiselle, 2019*; *Dyment et al., 2014*; *Dyment et al., 2013*; *Howell et al., 2017*; *Sakabe et al., 2018*). Interestingly, in aging, tendon cell number per unit area decreases, suggesting declining proliferative abilities with age (*Dunkman et al., 2013*). Consistent with this interpretation, tendon-derived cells from aged human tendons have reduced proliferative abilities and increased markers of senescence compared with adult-derived tendon cells in culture (*Kohler et al., 2013*). Together, these studies indirectly indicate that adult and aged tendon cells have reduced proliferative activity, yet sub-populations of tendon cells may divide in adult tendons under specific injury conditions. Nevertheless, it remains unclear to what extent, if any, there is physiological cell turnover in the adult tendon without injury and how this may differ from cell turnover during periods of active tendon growth.

Therefore, we sought to examine cell turnover rates in limb tendons during growth, adulthood, and aging. Using complementary methods of genetic pulse-chase labeling to trace the cell division history and BrdU/EdU incorporation to detect proliferation, we were able to identify changes in tendon cell turnover from birth to the early juvenile period (beginning around 3–4 weeks), with comparisons to adult and aged stages ($\geq$3 months and $\geq$18 months, respectively). We detect relatively high levels of proliferation during the neonatal period (P0-P7) and a rapid decline by P21. Although proliferation was significantly reduced after one month of age, surprisingly we were able to identify a small population of tendon cells that continued to proliferate in adult and aged mice, albeit at a very low rate. Understanding which cell populations can continue to divide in adults and the mechanisms driving the switch from proliferative to more quiescent stages would greatly benefit clinical approaches to tendon injuries.

## Results

### H2B-GFP pulse chase experiments demonstrate a shift from high to low proliferation rates in postnatal mice

To characterize cell proliferation in the tendon, we used the doxycycline (Dox) inducible Histone 2B-green fluorescent protein reporter mouse model (*Col1a1-tetO-H2B-GFP*; *ROSA-rtTA*, henceforth referred to as H2B-GFP), which has been used to quantify cell proliferation and identify slowly cycling label-retaining cell populations based on the stability and dilution of H2B-GFP protein in each cell (*Chakkalakal et al., 2014*; *Foudi et al., 2009*). After H2B-GFP expression is induced by Dox addition, Dox is removed for the chase period and H2B-GFP protein becomes diluted in proportion with each subsequent cell division (*Figure 1A*). Therefore, cells cycling more frequently will dilute H2B-GFP protein more quickly and will appear unlabeled earlier in the chase period; more slowly cycling cells will retain H2B-GFP protein longer during the chase. To verify the H2B-GFP system worked efficiently in the tendon, we pulsed mice with Dox from embryonic stage (E) 10 to birth and examined H2B-GFP expression on postnatal day (P) 0. Two photon microscopy images of histological sections of pulsed P0 Achilles (*Figure 1B–B''*) tendons showed widespread expression of H2B-GFP throughout the Hoechst$^+$ tendon nuclei. We next confirmed by flow cytometry that more than 90% of the tendon cells were positive for H2B-GFP at P0 (*Figure 1D*), indicating efficient labeling of all tendon cell populations examined. To ensure we were enriching for tendon cells, we only analyzed cells from dissected tendon tissues that were negative for CD45 and CD31 to remove blood and endothelial cells, respectively (*Figure 1—figure supplement 1A*). We also found the background level of H2B-GFP expression without the addition of Dox was very low (<1%; *Figure 1—figure supplement 1B*). At all stages analyzed, tendon cells were isolated from extensor, deep and superficial flexor, and Achilles tendons in the hindlimbs and extensor, deep and superficial flexor tendons in the forelimbs.

To determine the total cumulative proliferation of tendon cells from birth to aged mice, we next examined H2B-GFP expression in tendons that had been pulsed with Dox at embryonic stages and allowed to chase without Dox for over 18 months using section and FACS analysis. Tendons in section imaged using 2-photon microscopy appeared to have reduced H2B-GFP$^+$ expression in Hoechst$^+$ nuclei (*Figure 1C–C''*) compared to P0 tendons (*Figure 1B–B''*). We found that H2B-GFP$^+$ cells had shifted in the intensity of GFP (*Figure 1E*) with only 20.1 ± 1.4% of the cells H2B-GFP$^+$ at 645 days (*Figure 2A*). Previous studies calculate that 7-8 divisions are needed for a cell to fall below the GFP detection threshold (*Foudi et al., 2009*). This would indicate that the H2B-GFP$^+$ population

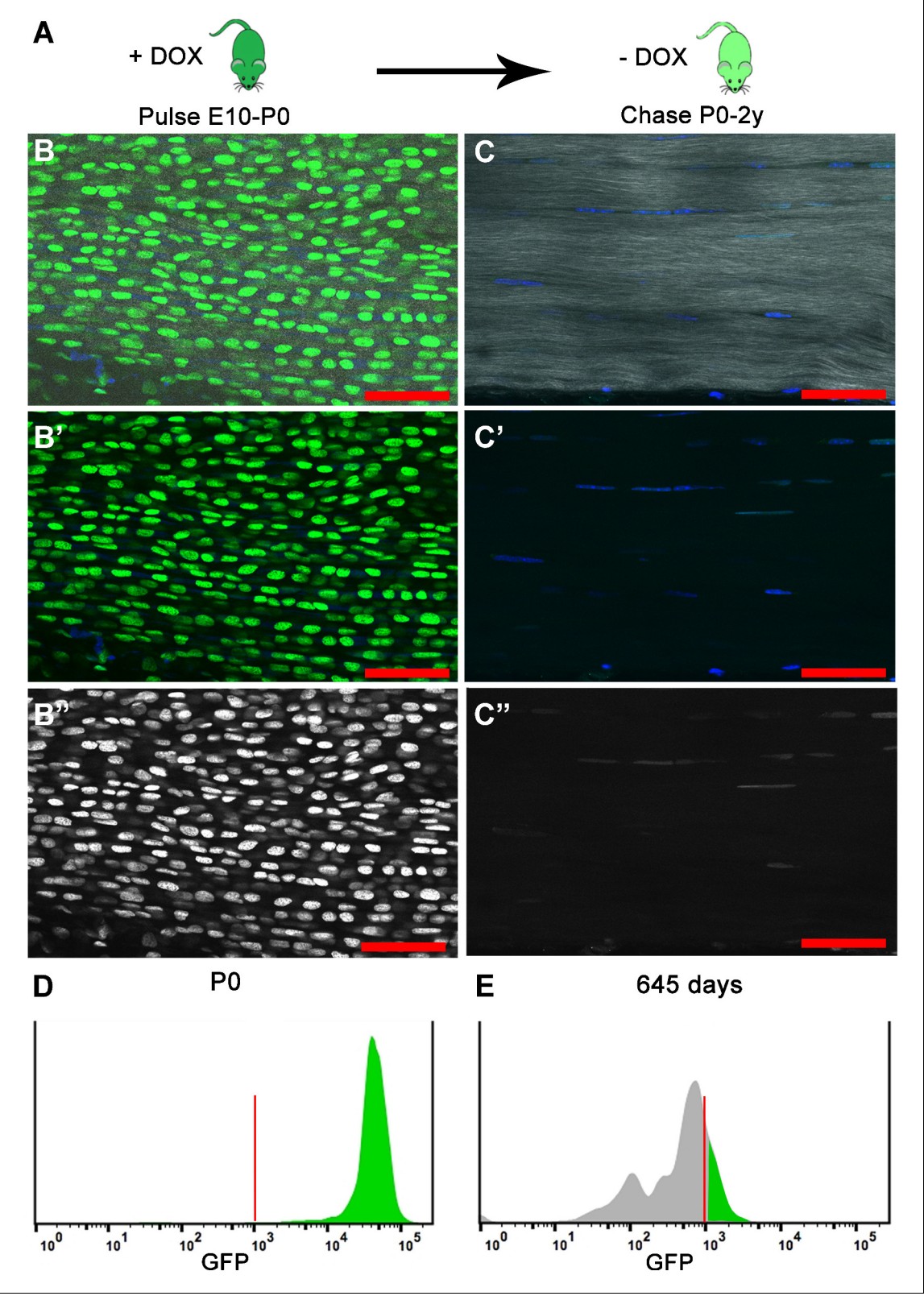

**Figure 1.** H2B-GFP expression is induced upon the addition of Dox to timed pregnant females from E10 to P0 (dark green); Dox is removed for the chase period of 0–2 years and H2B-GFP protein is diluted (light green) in proportion to cell division. (A) At birth (P0), longitudinal sections of Achilles (B–B'') tendons (n = 3 mice) show extensive H2B-GFP⁺ (green B, B'; white B'') labeling of Hoechst⁺ nuclei (blue, B, B'). SHG is shown in white (B, C). Histogram showing that more than 95% of the cells are H2B-GFP⁺ at P0 (D). After 680 days, Achilles (C–C'') tendons (n = 3 mice) have qualitatively

*Figure 1 continued on next page*

*Figure 1 continued*

fewer H2B-GFP[+] (green C, **C'**; white **C''**) labeled Hoechst[+] nuclei (blue C, **C'**). Histogram showing only 20% of the cells are H2B-GFP[+] and the H2B-GFP intensity has decreased at 645 days (**E**). For the histograms, a representative is shown; tendons from n > 3 mice were examined independently. Scale Bars, 50 µm; Vertical red lines (**H, O**) indicate the control GFP beads for standardizing intensity and gates.

DOI: https://doi.org/10.7554/eLife.48689.003

The following figure supplement is available for figure 1:

**Figure supplement 1.** Flow cytometry analysis of H2B-GFP[+] tendon cells showing the exclusion of CD31[+] and CD45[+] cells; CD31[-]/CD45[-] cells (80–90%) were used for our H2B-GFP analysis.

DOI: https://doi.org/10.7554/eLife.48689.004

at 645 days proliferated less than 7-8 times, while the H2B-GFP[−] population proliferated at a minimum of 7-8 times since birth. Together, these data show that all tendon cells proliferate after birth, but that a subpopulation of the cells display limited proliferative activity.

To more deeply assess the dynamic changes of tendon cell proliferation after birth, we analyzed Dox pulsed mice at multiple stages of chase from P0 to P80. We observed a decrease in the total percentage of H2B-GFP[+] cells from 93.6 ± 1.2% at P0 to 76.7 ± 9.6% at P7% and 52.0% at P14 (**Figure 2A**). However, the percentage of H2B-GFP[+] tendon cells remained relatively constant between P14 and P80 with no significant differences among any pair of time points (**Figure 2A**), suggesting limited cell division occurred from P21-P80. Interestingly, the percentage of H2B-GFP[+] cells was further reduced at 645 days to 20.1 ± 1.4% indicating low but detectable amounts of cell proliferation continue in adult and aged mice (**Figure 2A,B**).

As significant changes in the percentage of positive and negative H2B-GFP cells between P14-P80 were not observed, we next examined alterations in H2B-GFP intensity, as this would reveal more subtle changes in cell division that occur. We noted a marked shift in the H2B-GFP[+] intensity from $10^5$ at P0 to $10^3$ after 645 days (**Figures 1E** and **2C**). Using a logarithmic decay equation to define the dilution of GFP signal mathematically (see Materials and methods), we also observed increased proliferation at early stages (**Figure 2D**). Our calculations show that tendon cells were dividing at a rate of 19 ± 4.2% per day from P0 to P7 and 9 ± 4.25% per day from P7 to P14 (**Figure 2D**). Proliferation rates decreased to 3.85 ± 0.07% per day between P14 and P21 and 1.75 ± 0.64% per day from P21 to P80. After P80, the rate of tendon cell proliferation was markedly decreased to 0.1 ± 0.13% per day by P600 (**Figure 2D**). Together, these proliferation rates derived from mathematical modeling of H2B-GFP decay and the absolute loss of H2B-GFP intensity over time from our flow cytometry analysis indicate that there are relatively high levels of proliferation at the early postnatal stages. In addition, this proliferative activity is greatly diminished after one month of age, but not extinguished in adult or aged tendons.

## BrdU incorporation analysis identifies a postnatal transition from high to low cell division rates

To complement our mathematical model of H2B-GFP decay, we used flow cytometry to quantify the percentage of tendon cells that had incorporated Bromodeoxyuridine (BrdU), a thymidine analog that incorporates into replicating DNA, for different BrdU administration lengths and stages. We performed intraperitoneal (IP) injection of BrdU and harvested tendons to determine the number of BrdU[+] cells after 24 hours. For flow cytometry analysis, highly proliferative organs (gastrocnemius muscle) were used as positive controls, tendon tissues from mice that had not received BrdU treatment were used as negative controls, and *Scleraxis (Scx)-Cre;Rosa-LSL-TdTomato[+]* (abbreviated *Scx-Cre;TdTom*) mice were used to analyze *Scx*-descendent tendon cells (*Blitz et al., 2009*). We found that BrdU injection at P0 resulted in 76 ± 13.8% BrdU[+] tendon cells at P1, while at P8 and P22, 26 ± 6.5% and 8.4 ± 3.4% of the tendon cells were BrdU[+] positive, respectively (**Figure 3A,B**). In adult mice, we observed that less than 1% of the cells were BrdU[+] (P60 = 0.4 ± 0.2%, P370 = 0.5 ± 0.1%). To verify these findings in tissue sections, we injected EdU at P1 and P59 and examined *Scx-Cre;TdTom[+]* and EdU[+] tendon cells in section one day later. Consistent with our BrdU and H2B-GFP results, we observed more *Scx-Cre;TdTom[+]*/EdU[+] cells in the Achilles tendon at P2 compared with P60 mice (**Figure 3C**). Interestingly, we also observed noticeable doublets of EdU[+] cells in rows along on the longitudinal axis of the Achilles tendon (**Figure 3C,B'**). This indicates that the cells divided and retained their relative position in channels along the long axis of the tendon and is

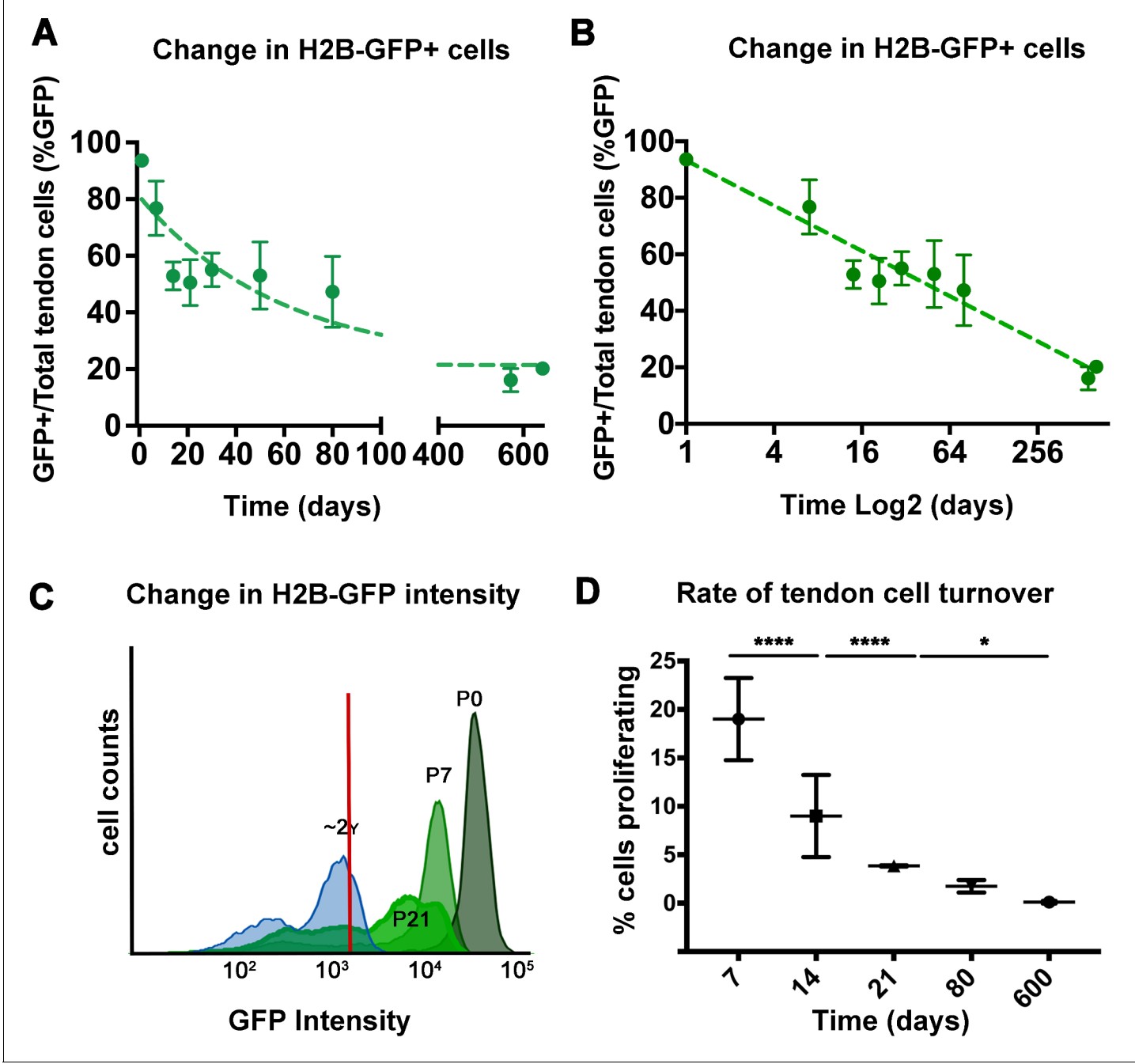

**Figure 2.** Analysis of H2B-GFP+ cells from postnatal to aged stages. The percentage of H2B-GFP+ tendon cells decreases from P0 to P645, with a rapid decline of H2B-GFP+ tendon cells by 14 days post birth and a more graduate decrease in H2B-GFP+ cells from P14 to P645 (**A**). Analysis of H2B-GFP+ tendon cells shows logarithmic decay (log2) in the percentage of H2B-GFP+ cells (**B**). H2B-GFP+ tendon cells decrease in their intensity of GFP from P0 to P645 (**C**; stages examined are labeled and shown with different shades of green; histogram is a representative from n = 4 mice). Using the change in H2B-GFP intensity, a model was used to calculate the daily cell proliferation rates (**D**), which shows significantly higher proliferation rates from P0 to P21, and lower proliferation rates at P80 and P600. Tendons from n > 3 mice were examined per each stage shown on the graph. All error bars represent standard deviation.

DOI: https://doi.org/10.7554/eLife.48689.005

The following source data is available for figure 2:

**Source data 1.** Source data for *Figures 2A–C* and *3A-D*.
DOI: https://doi.org/10.7554/eLife.48689.006
**Source data 2.** Source data for *Figure 2D*.
DOI: https://doi.org/10.7554/eLife.48689.007

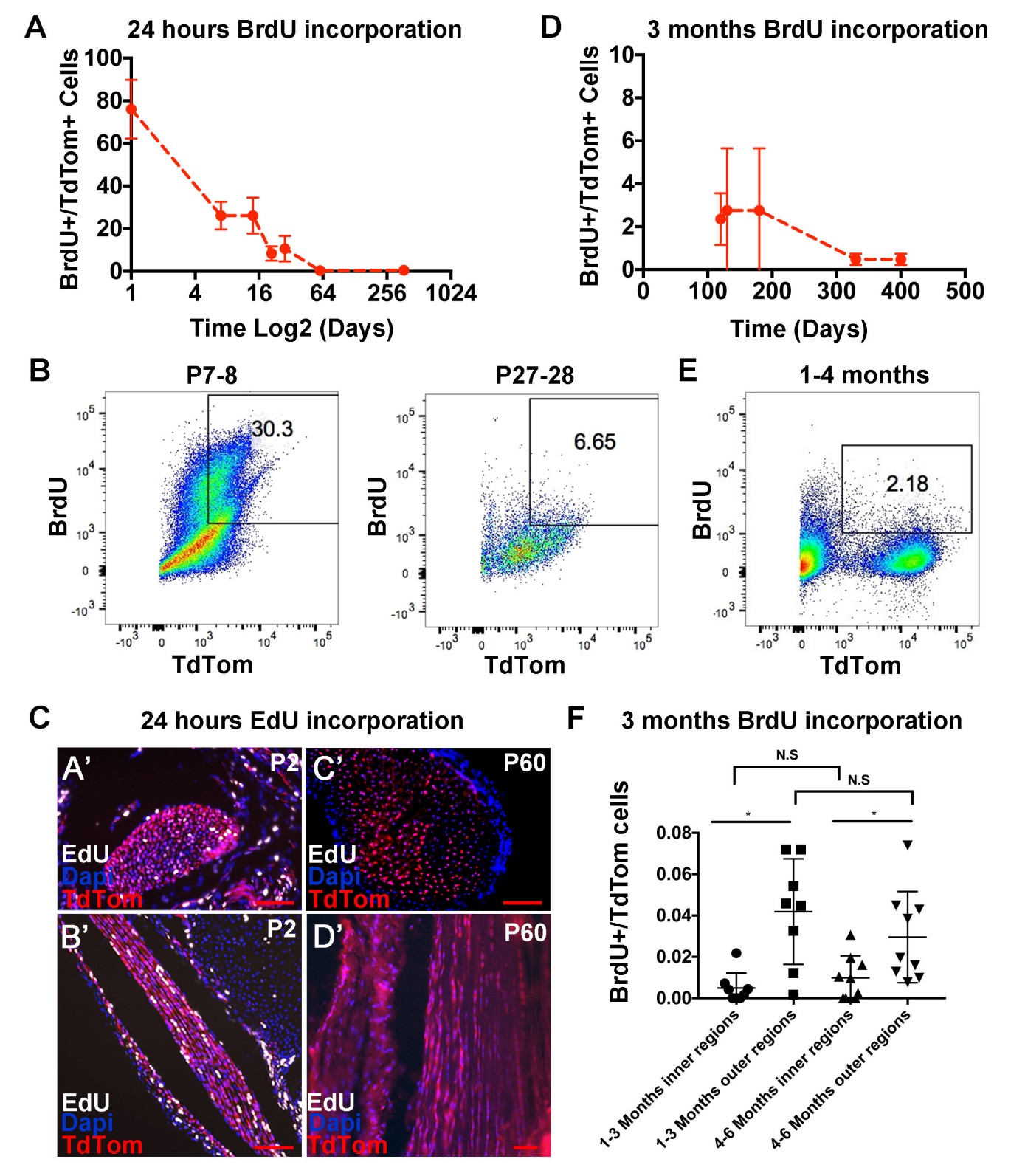

**Figure 3.** Analysis of tendon cell proliferation using short and long pulses of BrdU/EdU. Examination of BrdU incorporation 24 hr after BrdU injection reveals higher BrdU incorporation rates at P0 through P14 compared with P21 and later stages (A). Representative flow cytometry of 24 hr BrdU labeled *Scx-Cre;TdTom*+ tendon cells shows 30.3% BrdU+/TdTom+ cells at P8 and 6.65% BrdU+/TdTom+ cells at P28 (B). 24 hr EdU labeling identifies a significant number of proliferating cells at P2 (C, A', B'), but no EdU+ cells were observed at P60 (C, C',D'). Administration of BrdU for over 90 days

*Figure 3 continued on next page*

Figure 3 continued

after P30 shows low but detectable levels of BrdU incorporation from 120 to 400 days (D). Example flow cytometry shows 2.18% BrdU$^+$/TdTom$^+$ cells after 3 months of BrdU incorporation (1–4 months, (E). Quantification of BrdU$^+$/TdTom$^+$ cells in tendon sections (F) shows a similar percentage of incorporation as the flow cytometry analysis and significantly more BrdU incorporating nuclei in outer Achilles tendon regions compared with inner regions at either 3 and 6 months. For all experiments, n = 3 mice were examined per stage. Error bars are standard deviation. Scale bars = 100 μm.
DOI: https://doi.org/10.7554/eLife.48689.008

The following source data and figure supplement are available for figure 3:

**Source data 1.** Source data for *Figure 3F*.
DOI: https://doi.org/10.7554/eLife.48689.010

**Figure supplement 1.** No significant differences in the percentage of BrdU$^+$ cells were observed between *Scx-Cre;TdTom$^+$* cells and *Scx-GFP$^+$* cells after 120 days of BrdU incorporation.
DOI: https://doi.org/10.7554/eLife.48689.009

consistent with prior work noting an increase in cell number along the longitudinal axis at postnatal stages (*Kalson et al., 2015*). These results show a high rate of proliferation immediately following birth, and a decrease in the first weeks of postnatal life, specifically after P21, which is consistent with our H2B-GFP mathematical model. However, the low percentage of BrdU$^+$ cells at P60 and P370 suggests minimal turnover in adult tendons. To more accurately quantify the amount of cell proliferation in adults, we administered BrdU continuously in the drinking water of *Scx-GFP;Scx-Cre; TdTom* mice for 90 to 100 days. We found that after long periods of BrdU administration, 4 month old mice had incorporated BrdU into 2.35 ± 1.2% of the *Scx-Cre;TdTom$^+$* cells and 2.75 ± 2.9% of the *Scx-GFP$^+$* cells (*Figure 3D,E*; *Figure 3—figure supplement 1*), using flow cytometry. Quantification of BrdU stained tendon sections at 3 and 6 months showed a similar percentage of BrdU$^+$ cells (*Figure 3F*), further supporting a low, but detectable rate of turnover in adult mouse tendons. We also observed a significantly greater number of cells incorporating BrdU in the outer compared with internal tendon regions (*Figure 3F*). In mice older than 1 year of age, 90 days administration of BrdU yielded 0.48 ± 0.26% of BrdU$^+$ tendon cells (*Figure 3D*), however, this decrease was not statistically significant between 4, 6 and 13 month stages.

## Dynamic gene expression changes occur during the transition in cell division rate

Since we have determined that there is a transition in cell division rate during the first postnatal month, we also predict that there are dynamic gene expression changes occurring during this period, especially for genes important for proliferation and matrix production. We performed RT-qPCR assays on RNA isolated from whole distal limb tendon homogenate for a small set of transcripts. These assays provide further information about cell proliferation (*Mki67*), tendon cell identity and differentiation (*Scx, Mkx*), and matrix production and assembly (*Col1a2, Col3a1, Fmod*), during tendon growth. An analysis of variance (ANOVA) on ΔC$_T$ values for each gene demonstrated a significant change in expression of all genes across the developmental range (p<0.05). Tukey's Honestly Significant Difference (HSD) *post hoc* tests revealed the specific pairs of time points for which relative expression is significantly different (see *Supplementary file 1*).

For many of the genes, relative expression levels decreased during the first month of age. Although KI-67 protein expression is commonly used as a marker of proliferating cells, *Mki67* mRNA expression has been shown to correlate with protein levels and the number of KI-67 positive cells seen in histological sections (*Prihantono et al., 2017.*; *Schleifman et al., 2014*). Based on this, we examined *Mki67* transcript levels as another independent way to assess the number of mitotically active cells. *Mki67* gene expression was highest during the first week after birth (P0 to P7), and no significant differences were observed between P0, P7, and P14 (all p>0.8; *Figure 4*; *Supplementary file 1*). By P21, however, the relative amount of *Mki67* mRNA present in the tendon became significantly reduced compared to earlier timepoints (P0, P7, and P14, all p<0.05; *Supplementary file 1*) and remained low throughout the rest of the time series. By P35, *Mki67* expression levels approached the lower limit of detection for our RT-qPCR assays (C$_T$ values ~35). Therefore, these results suggest that the number of proliferating cells is highest during the first week after birth, but by P35 most tendon cells are no longer mitotically active. The expression of *Scx, Mkx, Col1a2,* and *Col3a1* measured via RT-qPCR also decreased by P35 compared with P0,

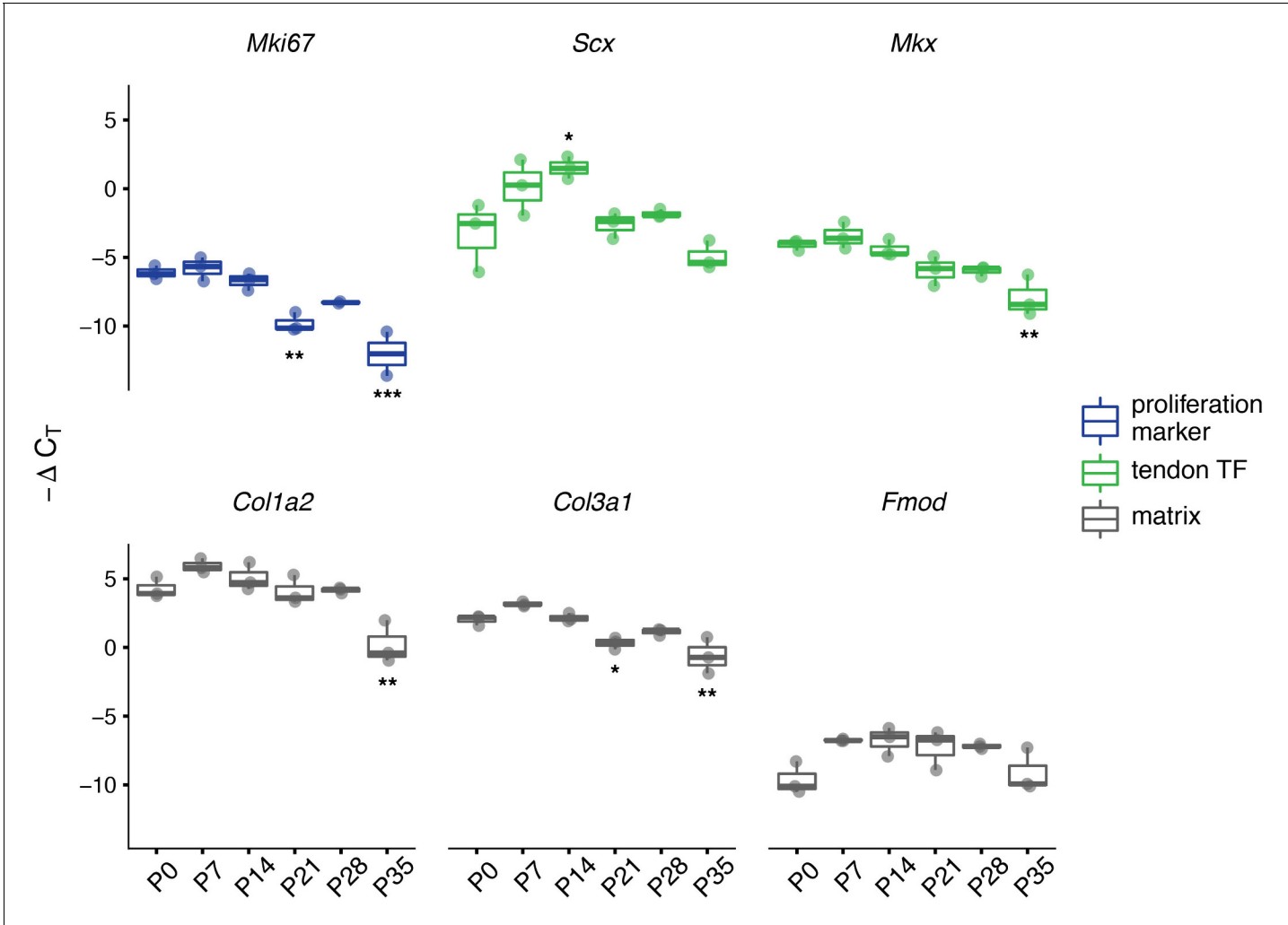

**Figure 4.** Expression of tendon and matrix related genes changes during the transition in cell division rate. RT-qPCR of selected markers of proliferation (*Mki67*; blue), tendon transcription factors (TFs, *Scx* and *Mkx*; green), and extracellular matrix (*Col1a2, Col3a1,* and *Fmod*; gray). Relative expression was calculated using the $\Delta C_T$ method using *Gapdh* as the reference gene. For all genes assayed, significant differences between all six time points were found via ANOVA ($p<0.05$). Stars indicate significant differences based on Tukey's HSD compared to P0 only (*$p<0.05$; **$p<0.01$; ***$p<0.001$). See **Supplementary file 1** for ANOVA statistics and full report of *post hoc* pairwise comparisons. n = 3 biological replicates per time point. Boxplot edges represent the interquartile range (IQR) and the middle line represents the median. Whiskers represent 1.5 x IQR.
DOI: https://doi.org/10.7554/eLife.48689.011

while *Scx* alone shows significantly increased expression at P14 relative to birth and later stages (*Figure 4*; *Supplementary file 1*). *Fmod* expression follows a different pattern, however, with higher transcript measurements at all timepoints from P7 to P28 compared to P0; however, none of these differences in *Fmod* expression achieved statistical significance during *post hoc* testing (*Figure 4*; *Supplementary file 1*).

## Tendon cell density and tendon length undergo dynamic changes during early postnatal stages

To understand how tendon cell number changes relative to matrix expansion during growth, we also quantified tendon cell density during the first postnatal month. Using 2-photon microscopy and second harmonic generation (SHG) imaging to generate 3D images of *Scx-GFP*[+] Achilles tendons, we counted Hoechst[+] cells and examined collagen organization at P0, P7, P14, and P28 (*Figure 5A–D'''*; *Videos 1–5*). As has been previously reported (*Kalson et al., 2015*), we observed a decrease in cell density per unit area, with 42.3 ± 11.44, 26 ± 4.6, 21.8 ± 3.3, and 9.6 ± 1.9 cells per 50 mm x 50

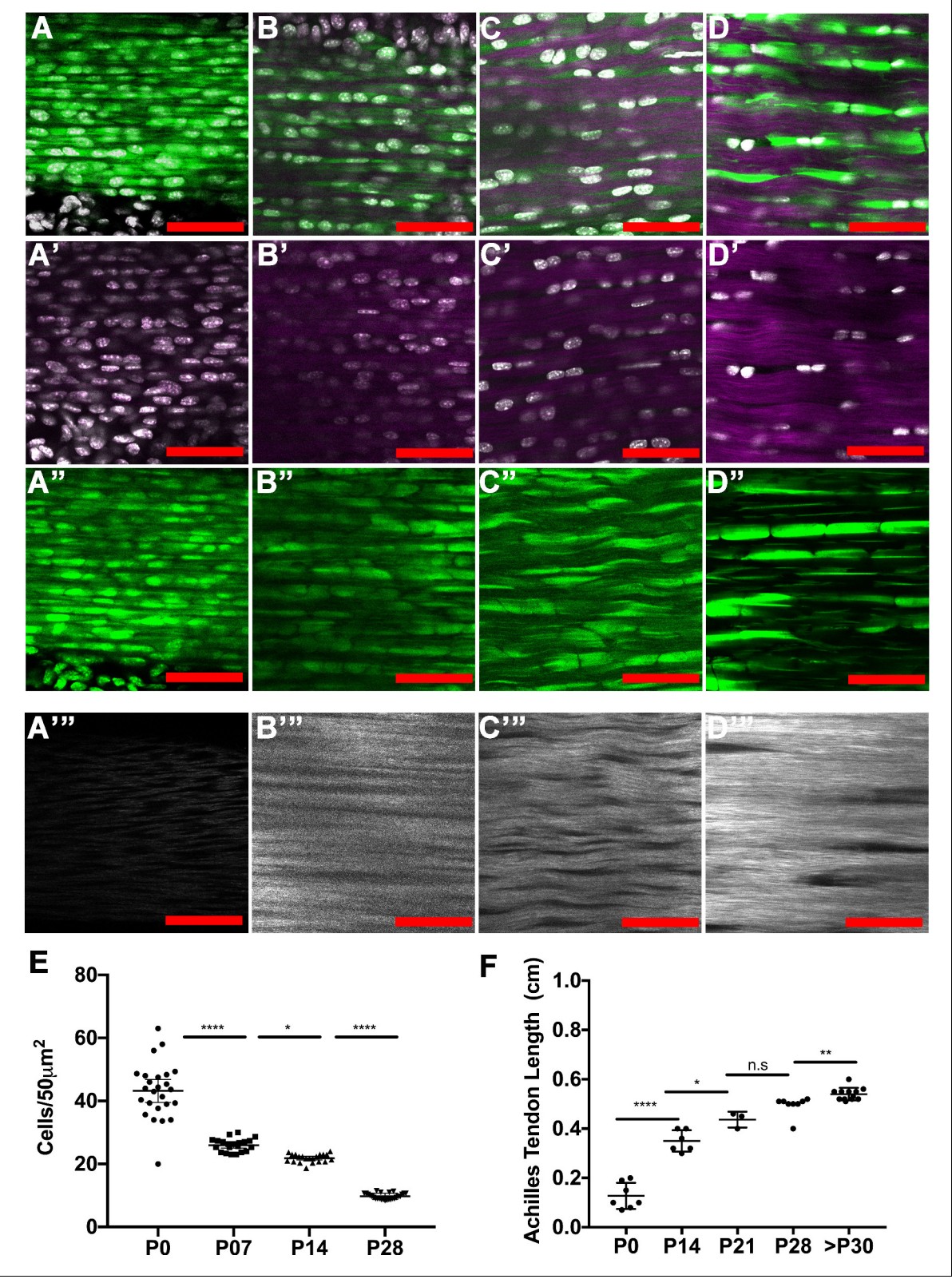

**Figure 5.** Tendon growth corresponds to changes in tendon cell density and matrix organization. Two photon microscopy images of Achilles tendons showing second harmonic generation signal (purple, A-D'; white, A'''–D'''), Hoechst+ nuclei (white, A–D'), and *Scx-GFP*+ cells (green, A–D, A''–D''). Images show changes in cell density, cell shape, and collagen organization at P0 (A–A'''), P7 (B–B'''), P14 (C–C'''), and P28 (D–D'''). Using optical sections from 2-photon images, we found that tendon cell density significantly decreases at postnatal stages (E, each point represents cell counts from

*Figure 5 continued on next page*

*Figure 5 continued*

an optical section; n = 3 mice analyzed per stage). The Achilles tendon undergoes significant longitudinal growth from P0 to P30; no significant increases in Achilles length is detected after P30 (F, each point represents measurements from one mouse Achilles; n > 3 mice). Error bars are standard deviation. Scale bars = 100 μm.

DOI: https://doi.org/10.7554/eLife.48689.012

The following source data is available for figure 5:

**Source data 1.** Source data for *Figure 5A*.

DOI: https://doi.org/10.7554/eLife.48689.013

**Source data 2.** Source data for *Figure 5F*.

DOI: https://doi.org/10.7554/eLife.48689.014

mm at P0, P7, P14, and P28, respectively (*Figure 5E*). These results suggest that matrix expansion outpaces cell proliferation, at least for the cross-sectional area of the tendon. Consistent with this observation, using the same imaging conditions, we observed an increase in SHG signal intensity from P0 to P28, suggesting an increase in collagen density at these stages (*Figure 5A'''–D'''*). We also noticed a larger variability in cell density at P0 compared to P28, which could indicate natural variability in growth rates during early stages. To understand how cell division compares with longitudinal tendon growth, we measured the Achilles tendon length from the enthesis to its connection with the gastrocnemius muscle at postnatal and adult stages. Strikingly, we observed rapid growth in the early postnatal stages with the Achilles tendon increasing from 0.127 ± 0.019 cm at P0 to 0.35 ± 0.017 cm at P14, and to 0.436 ± 0.018 cm at P21. However, the length of the Achilles tendon did not change significantly between P21 to P28 (*Figure 5B*), and only increased modestly from 0.496 ± 0.01 cm at P28 to 0.54 ± 0.007 cm after P30 (P30-P270) (*Figure 5B*). Overall, the time periods where we observed significant increases in Achilles tendon length correspond directly with our observations of periods of active tendon cell proliferation. This suggests the interesting possibility that, in parallel with matrix expansion, cell proliferation during the first two weeks after birth may in some way contribute to longitudinal growth or result from mechanical or chemical changes that occur during this dynamic longitudinal growth period.

## Discussion

Defining the transition from developmental growth to adult homeostasis is important for understanding functional tissue physiology. Adult tissues range from high self-renewal activity driven by stem cell populations, such as in the blood and intestine, to low or even no self-renewal as has been reported for the liver and heart, respectively. The tendon presents an intriguing case as growth and maintenance involve both its highly organized matrix and the cells that reside within it. Many studies have highlighted the changes that tendon matrix undergoes in growth, adulthood, and aging. However, the activity of the cells as the matrix transitions from growth to maturation is less well understood. Previous work has suggested that cell proliferation in adult tendons is limited (*Runesson et al., 2013*), but it is unclear when and to what extent this decline in cell division occurs. Identifying the shift from proliferative growth to homeostasis is also important for properly defining cellular growth periods and understanding self-renewal mechanisms in the adult tendon.

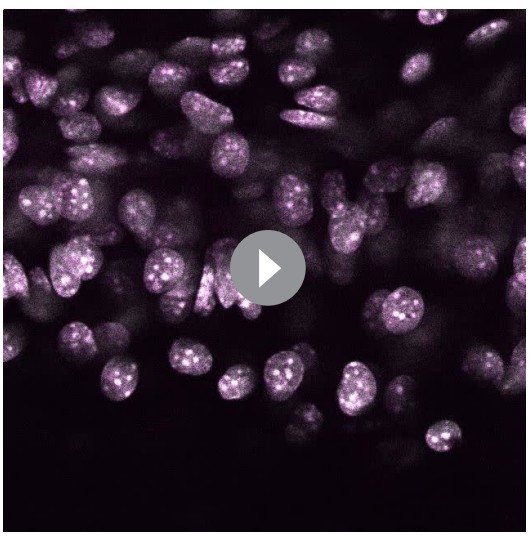

**Video 1.** Movie of sagittal optical sections taken using 2-photon microscopy of P0 Achilles tendon with Hoechst[+] nuclei (white) and SHG signal (purple) corresponding to *Figure 5A'*.

DOI: https://doi.org/10.7554/eLife.48689.015

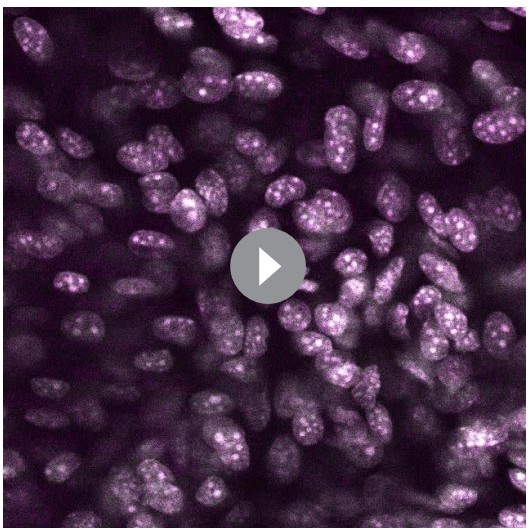

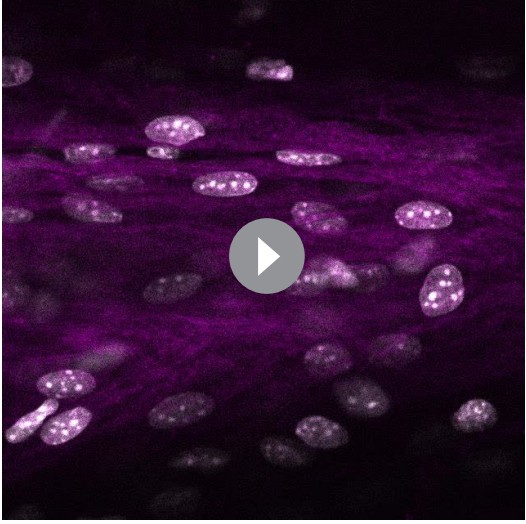

**Video 2.** Movie of sagittal optical sections taken using 2-photon microscopy of P7 Achilles tendon with Hoechst[+] nuclei (white) and SHG signal (purple) corresponding to *Figure 5B'*.
DOI: https://doi.org/10.7554/eLife.48689.016

**Video 3.** Movie of sagittal optical sections taken using 2-photon microscopy of P14 Achilles tendon with Hoechst[+] nuclei (white) and SHG signal (purple) corresponding to *Figure 5C'*.
DOI: https://doi.org/10.7554/eLife.48689.017

During postnatal development, the tendon ECM undergoes increases in collagen fibril diameter, collagen content, and mechanical properties (*Ansorge et al., 2011*). In our study, we sought to define the changes that occur to the cells within the tendon during the same periods of growth and homeostasis. Using the H2B-GFP system and BrdU/EdU labeling, we detected significant cell proliferation prior to one month of age. In addition to loss of H2B-GFP, the intensity of H2B-GFP expression decreased demonstrating that all tendon cells divide at least once during postnatal life. The decrease in H2B-GFP intensity across the time series is best described by a logarithmic decay model, which yields proliferation rates similar to those measured via BrdU labeling. Although there were some discrepancies between BrdU labeling and our H2B-GFP mathematical model at P21, these differences were modest and could be attributed to differences in BrdU incorporation into the tendon, a low level (<1%) leakiness of the H2B-GFP system (*Figure 1—figure supplement 1B*), or a potential difference in cell populations as we examined *Scx*-lineage (*Scx-Cre;TdTom+*) cells with BrdU and total tendon cells negatively sorted for CD31/CD45 with H2B-GFP. Despite the potential drawbacks from each method, we obtained similar results from these complementary approaches further strengthening our conclusions. Future analysis focusing on specific subpopulations of cells residing in the tendon –such as macrophages, the sheath or epitenon cells, tendon-derived stem/progenitor cells (*Bi et al., 2007*), the S100a4-expressing population (*Best and Loiselle, 2019*), the αSMA-expressing cells (*Dyment et al., 2014*), or through the use of extracellular matrix reporters (reviewed in *Delgado Caceres et al., 2018*)– would further refine this analysis and determine if the mitotic potential in the growing and adult tendon is

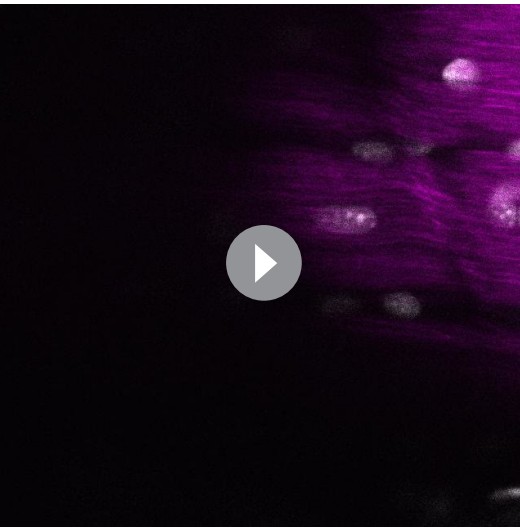

**Video 4.** Movie of sagittal optical sections taken using 2-photon microscopy of P28 Achilles tendon with Hoechst[+] nuclei (white) and SHG signal (purple) corresponding to *Figure 5D'*.
DOI: https://doi.org/10.7554/eLife.48689.018

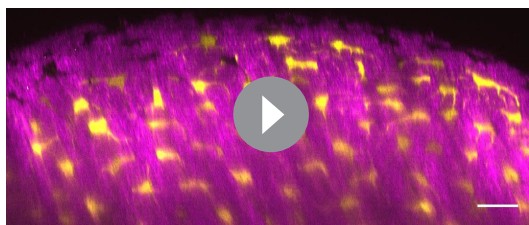

**Video 5.** Movie of optical sections taken using 2-photon microscopy of P28 Achilles tendon with *Scx-GFP*[+] cells (yellow) and SHG signal (purple). For this movie, the 'reslice' feature in the FIJI software was used to convert the image from the sagittal to transverse view.

DOI: https://doi.org/10.7554/eLife.48689.019

restricted to specific populations of tendon cells. In relation to the growth of the tissue and consistent with others (*Dunkman et al., 2013*; *Kalson et al., 2015*), we have observed decreased cell density and an increase in collagen organization as observed by second harmonic generation signal in postnatal Achilles tendons as the mice mature from P0 to P28. This corresponds with rapid elongation of the Achilles tendon in the early postnatal stages with little change occurring from P30 to adulthood. Taken together, our analyses show there is significant proliferation even as the tendon cells are reduced in density. Although this indicates that matrix expansion outpaces cell growth, it also points towards a possible co-regulation of proliferation and matrix expansion during early postnatal stages, which could have interesting implications for how cells regulate, or respond to, ECM expansion and changes in biochemical and mechanical signals.

Our gene expression analysis also demonstrates interesting changes in the first month of this transition from growth to homeostasis. Relative expression of *Mki67* is significantly downregulated by P21 compared to the earlier time points. In later stages of the time series *Mki67* transcripts are reduced to nearly undetectable levels ($C_T$ ~35). Concurrently, the relative expression patterns of tendon transcription factors (*Scx* and *Mkx*), and pro-collagen genes (*Col1a2* and *Col3a1*) largely match that of *Mki67*. Both Scx (*Murchison et al., 2007*; *Schweitzer et al., 2001*; *Shukunami et al., 2006*) and Mkx (*Ito et al., 2010*; *Liu et al., 2014*) are involved in tenocyte differentiation, as well as matrix organization via interactions with Smad3 (*Berthet et al., 2013*). Our findings on the coordinated downregulation of *Scx, Mkx, Col1a2,* and *Col3a1* after P14 fits within this established framework and suggests that the period from P0 to P14 is a key window of postnatal tendon development. The persistence of *Fmod* expression is concordant with previous studies of ECM proteins during the postnatal period in mice (*Ezura et al., 2000*), indicating that collagen fibril formation slows early, but fibril growth, mediated by *Fmod*, continues into the juvenile period (>1 month). Although our gene expression analysis is consistent with previous studies, this work was limited to a handful of tendon genes and an unbiased next generation sequencing method such as RNA-seq would provide a more comprehensive analysis of gene expression during this postnatal transition.

The tendon has also been shown to undergo regenerative healing during fetal and early postnatal periods (*Ansorge et al., 2011*; *Favata et al., 2006*; *Howell et al., 2017*). The timing in these studies is reminiscent of the other organ systems such as the heart (*Bassat et al., 2017*), which demonstrate more regenerative potential at neonatal compared to adult stages. In mice, tendons injured prior to one week of life undergo regenerative healing, with mechanical properties of the healed tendon nearly matching those of the uninjured controls; injured tendons of mice older than 3 weeks of age healed imperfectly through scar formation (*Ansorge et al., 2011*; *Howell et al., 2017*). Previous work has also demonstrated that the regenerative abilities of injured fetal sheep tendons are not affected by transplantation into an adult environment (*Favata et al., 2006*), suggesting that the regenerative properties of developing tendons are intrinsic. Interestingly, neonatal cardiac regeneration has been attributed to the ability of cardiomyocytes to proliferate during the first 1–2 weeks of postnatal life (*Bassat et al., 2017*). It is interesting to speculate that the swift decline in tendon cell proliferation that we observed at 3 weeks of age may also underlie the shift in regenerative to reparative healing in the tendon.

In addition to defining distinct postnatal periods of cell proliferation, our work also establishes the presence of cell division in tendon cells at adult and aged stages. One caveat of our study is that although mice over 18 months are considered "aged", age-related tendon phenotypes are not observed until after 22 months (*Ackerman et al., 2017*). It would be interesting to perform our long BrdU administrations at these stages to determine if significant differences in mitotic activity are observed. Despite the low levels of proliferation, we detected BrdU incorporation in both *Scx*-lineage and *Scx-GFP*[+] cells in adults. Although our current understanding of the self-renewal

mechanisms in the tendon are limited, studies have shown that tendon-derived stem/progenitor cells divide readily and are multipotent when isolated and expanded in culture (*Bi et al., 2007*). These cells can also form tendon-like tissues upon transplantation (*Bi et al., 2007*), but how this activity reflects that of resident cells in their native environment is unclear. In the context of injury, recent studies have shown contributions to the healing tissue from *Scx-GFP*-negative cells originating from tendon sheath regions (*Dyment et al., 2014*; *Wang et al., 2017*). These results are consistent with earlier studies showing that external tendon cell populations exhibited increased proliferative capacity compared with internal tendon cells in culture (*Banes et al., 1988*). Other studies have shown contributions from *Scx*- or *S100a4*-lineage cells to the bridging tendon tissue (*Best and Loiselle, 2019*), which would suggest that tendon cells within the tendon body are also capable of proliferating. However, it is unknown if these previously identified cells are responsible for the homeostatic proliferation detected in adults. A combined study using genetic lineage tracing of sub-populations of tendon cells with the H2B-GFP or BrdU labeling system would address this question. In examining human tendon tissue, studies have used Carbon-14 (C14) isotope analysis to infer human tendon turnover rates based on of known changes in atmospheric C14 levels originating from atomic bomb tests. These studies show that the majority of the tendon core mass is formed by adolescence (*Heinemeier et al., 2013*). Consistent with this previous study, our results show that most cell division in the tendon occurs prior to the juvenile stage. However, our work also indicates continued low levels of proliferative activity in adults. As the previous C14 studies were performed with tissue samples, which are predominantly matrix, it is unclear, as the authors also note, if they could detect low rates of turnover by a small population of cells. Therefore, even though the C14 results indicate very little tissue turnover after adolescence, they do not exclude the possibility of a slowly cycling tendon cell population in humans. Interestingly, further C-14 analysis of collagen isolated from tendinopathy samples showed evidence of collagen turnover after adolescent periods (*Heinemeier et al., 2018*). Although it is unclear if the collagen turnover is a cause or effect of the tendinopathy, these results suggest that adult tendon cells can be actived at adult stages to remodel their matrix significantly.

In summary, by using complementary genetic and chemical labeling methods, we have gained a comprehensive understanding of the dynamic cell proliferation rates in the tendon from birth to aging. We show that limb tendon cells remain proliferative throughout early postnatal stages (P0-P21) and that mitotic activity declines significantly in juvenile periods with a small population of cells continuing to divide from one month and 1–2 years of life. The timing of these changes in cell turnover appears to be correlated with the timing at which the tendon matrix is undergoing expansion and maturation (*Figure 6*), as well as when the tendon cells are changing morphology from rounded to stellate. These changes in cell division also correlate with the transition from regenerative to

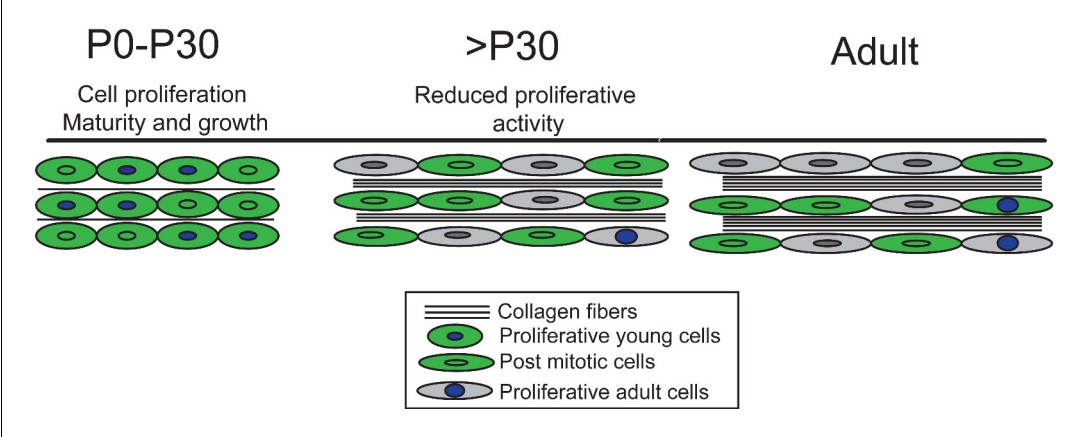

**Figure 6.** Our model of tendon growth shows significant proliferation prior to P30 that coincides with expansion and maturation of the tendon matrix. By P30, tendon cells transition from relatively high to very low rates of proliferation, signifying a transition from growth to physiological tissue maintenance. Others have shown that matrix maturation and expansion continue after P30, but these changes do not involve proliferation-driven cell growth or longitudinal growth of the tendon.
DOI: https://doi.org/10.7554/eLife.48689.020

reparative healing that has been documented in murine tendons. These findings are important to consider in studying tendon growth, maturation and self-renewal mechanisms, and have implications for identifying and characterizing self-renewal mechanisms of a tissue.

# Materials and methods

## Key resources table

| Reagent type (species) or resource | Designation | Source or reference | Identifiers | Additional information |
|---|---|---|---|---|
| Strain, strain background (*Mus musculus*) | Col1a1:tetO-H2B-GFP; ROSA:rtTA (H2B-GFP) | Brack, Hochedlinger labs, PMID: 24715455, 19060879 | | |
| Strain, strain background (*Mus musculus*) | Scx-GFP | Schweitzer lab, PMID: 20059955 | | |
| Strain, strain background (*Mus musculus*) | Scx-Cre | Schweitzer lab, PMID: 11585810 | | |
| Strain, strain background (*Mus musculus*) | Gt(ROSA)26Sortm9 (CAG.tdTomato)Hze (ai9)/TdTom | Jackson | cat# 007909 | |
| Antibody | Rat anti Mouse CD31 | BD | Cat#551262 | (1:100) |
| Antibody | Rat anti Mouse CD45 | BD | Cat# 557659 | (1:100) |
| Antibody | Mouse monoclonal BrdU | Biolegend | Cat# 339808 | (1:100) |
| Antibody | Rat monoclonal BrdU | Abcam | Cat# 6326 | (1:100) |
| Commercial assay or kit | Green Flow Cytometry Reference Beads | Molecular Probes | Cat# C16508 | |
| Commercial assay or kit | Click-iT EdU | Invitrogen | Cat# C10337 | |
| Commercial assay or kit | Qubit HS RNA assay | Invitrogen | Cat# Q32852 | |
| Commercial assay or kit | SuperScript | Thermo Fisher | Cat#18091050 | |
| Commercial assay or kit | SYBR | Applied Biosystems | Cat#4367659 | |
| Chemical compound, drug | Dox | Sigma | cat#D9891 | |
| Chemical compound, drug | Collagenase II | Worthington | Cat# LS004176 | |
| Chemical compound, drug | DMEM | Gibco | Cat#11956–092 | |
| Chemical compound, drug | P/S | Corning | Cat#30002 CL | |
| Chemical compound, drug | Hepes | Gibco | Cat#15630–80 | |
| Chemical compound, drug | Collagenase I | Gibco | Cat# 17100–017 | |
| Chemical compound, drug | Dispase | Gibco | Cat# 1710541 | |

*Continued on next page*

*Continued*

| Reagent type (species) or resource | Designation | Source or reference | Identifiers | Additional information |
|---|---|---|---|---|
| Chemical compound, drug | BrdU | Sigma | Cat#B5002 | |
| Chemical compound, drug | Hoechst | Thrmo Fisher | Cat# H3569 | |
| Chemical compound, drug | TRIzol | Invitrogen | Cat# 15596026 | |
| Software, algorithm | Prism8 | GraphPad | | |
| Software, algorithm | R | 'stats' version 3.5.1(Team) | | |

## Animals

We thank Andrew Brack (UCSF) and Konrad Hochedlinger (MGH) for the Doxycycline (Dox) inducible H2B-GFP (*Col1a1:tetO-H2B-GFP*; *ROSA:rtTA*) heterozygous mice used in these studies. To induce transgene expression, Dox (Sigma D9891, 2 mg/ml, supplemented with sucrose at 10 mg/ml) was added to the drinking water of timed pregnant females at E10-birth as described (*Foudi et al., 2009*). *Scx-GFP* and *Scx-Cre* mice were provided by the Schweitzer lab (*Blitz et al., 2009*; *Schweitzer et al., 2001*). Gt(ROSA)26Sortm9(CAG-tdTomato)Hze (*Ai9*) were obtained from Jackson Laboratory (Jax cat# 007909). All experiments were performed according to our protocol approved by the Massachusetts General Hospital Institutional Animal Care and Use Committee (IACUC: 2013N000062).

## Flow cytometry

The tendon cells were isolated from the distal forelimb and hindlimb tendon tissue (Achilles, extensor, deep and superficial flexor tendons) from mice at time points between P0 and 2 years. Limb tendons were enzymatically dissociated in a solution containing 0.2% collagenase II (Worthington Cat# LS004176) in DMEM (Gibco Cat#11956–092) with 1% P/S (Corning Cat#30002 CL) and 1% Hepes (Gibco Cat#15630–80) for 2 hr at 37˚C. Subsequently, a secondary digestion solution containing 0.2% Collagenase I (Gibco Cat# 17100–017) and 0.4% Dispase (Gibco Cat# 1710541) was added and the samples were incubated for an additional for 30 min at 37˚C. The digested cells were filtered with 30 µm filters (MACS Cat# 130041407) and washed. For the H2B-GFP$^+$ studies, we enriched for tendon cells from H2B-GFP$^+$ mice by excluding for CD31$^+$ and CD45$^+$ cells using FACS prior to analysis (BD Cat#551262, Cat# 557659). For the BrdU analysis, cells were stained with anti-BrdU following tendon tissue dissociation (Biolegend, Cat# 339808), and tendons from *Scx-Cre;TdTom$^+$*; or *Scx-Cre;TdTom$^+$;Scx-GFP$^+$* mice were used to analyze TdTom$^+$ or GFP$^+$ tendon cells. Flow cytometry was performed using 5 ml tubes (BD Biosciences Cat# 352235) on a FACSAria II (BD Biosciences). For each independent experiment, gates were defined by positive and negative control tendon cells from TdTom$^+$/TdTom$^-$ and GFP$^+$/GFP$^-$ cells. For the negative controls for BrdU analysis, BrdU antibody staining was performed on tendon cells isolated from mice that were not administered BrdU. To ensure reproducibility of H2B-GFP emission intensity between different samples and sorting times, the voltage of the photomultiplier receiving signal from the 488 nm laser was normalized using Green Flow Cytometry Reference Beads prior to every sort (Molecular Probes Cat# C16508).

## BrdU, EdU labeling, Tendon Histology, and Imaging

BrdU was injected at a concentration of 150 mg/kg (Sigma Cat#B5002) as described (*Magavi and Macklis, 2008*). Flow cytometry analysis was performed as described previously. For BrdU immunostaining, sections underwent antigen retrieval and immunostaining using anti-BrdU (1:100; Abcam Cat# 6326). EdU was administered at 20 mg/kg as described (*Salic and Mitchison, 2008*) and tendon sections were stained using the Click-iT EdU kit (Invitrogen Cat# C10337). For histological sections, tendons were fixed overnight in 4% PFA, followed by 5% sucrose for 1 hr, and 30% sucrose overnight before being mounted in OCT. A Leica cryostat (CM3050S) was used to obtain 8–10 µm

sections. Pictures were taken with Zeiss AxioImager D2 with (10X and 20X magnification) and prepared using Adobe Photoshop and Illustrator. For 2-photon imaging of H2B-GFP expression in mouse Achilles tendons (*Figure 1*), we chose 2-photon microscopy due to the longer wavelengths used to image the sample and the high collagen content of the tendon. The longer wavelengths penetrated the tendon tissue more efficiently and produced a better signal to noise ratio that was not affected by the collagen fibers, resulting in a more unified GFP signal. Achilles tendon samples from at least three mice were sectioned and analyzed at P0 (end of pulse) and 680 days (end of chase). For the P0-P28 data (*Figure 5*), at least 3 Achilles tendons from different mice were analyzed. For all 2-photon imaging, we stained the nucleus with Hoechst 33258 (ThermoFisher Cat# H3569) at 1:100,000 dilution. To standardize signal detection between sections and samples, the laser power was adjusted to 'Bright Z' mode. The images were analyzed using FIJI (*Schindelin et al., 2012*). For P28 transverse *Scx-GFP* Achilles tendon *Video 5*, the 'reslice' feature in the FIJI software was used to convert the image from the sagittal to transverse view (*Schindelin et al., 2012*). The images were taken with optical slices every 0.4 μm with a 25X wet lens (XLPlan N 25X WMP) on an Olympus 2P microscope FLOVIEW FVMPE-RS.

## Tendon length and cell counting measurements

Tendons from at least three mice were measured per stage from the calcaneus to the gastrocnemius muscle, and the data were analyzed using Prism software (Graphpad). For BrdU incorporation quantification (*Figure 3F*), we examined 3–4 tendon transverse sections in 3–4 regions along the tendon per mouse from at least three mice. In these transverse sections, we counted BrdU in the outer tendon regions within the outer 20 μm of the tendon and the inner regions comprised the regions internal from this area. The data points in *Figure 3F* represent the ratio of BrdU$^+$/TdTom$^+$ cells in the inner or outer regions of 3–4 sections from one region along the tendon. For cell density counting, we counted cells in three 50 μm x 50 μm squares in 7–9 optical transverse sections per Achilles tendon per mouse with a gap of 20 microns between sections and at least three mice were analyzed per stage. The 50 μm x 50 μm square was created using FIJI (*Schindelin et al., 2012*), and in each optical section the nucleus number was counted in at least three different locations.

## Mathematical modeling

To define the dilution of the GFP signal mathematically, we modeled the change in signal intensity using a logarithmic decay equation:

$$P(t) = P(0)e^{-kt}$$

In this formula, we assume that GFP signal intensity decreases through dilution by cell proliferation. We calculated the constant between populations at different times (k), by comparing the populations' median GFP intensity at particular times (P(t)). Assuming that the increase in tendon cell number could be measured by the decrease of the GFP intensity (*Figure 2B and D*), we calculated the dilution of GFP between each time point from P0 to 645 days (*Figure 2C*).

## RNA extraction and RT-qPCR

Fresh, whole limb tendons (pooled forelimb and hindlimb from a single individual; n = 3 mice per time point) were dissected from mice euthanized via $CO_2$ and immediately placed in cold TRIzol (Invitrogen 15596026). Tendons were roughly chopped with clean microdissection scissors in TRIzol and frozen at −80C until RNA extraction via TRIzol-chloroform and a proprietary kit. Briefly, the homogenate in TRIzol was thawed on ice, vortexed, and transferred to a clean microcentrifuge tube to remove tissue debris. The traditional TRIzol-chloroform extraction protocol was followed until phase separation. An equal volume of ethanol was added to the upper aqueous phase and the mixture was transferred to a Zymo IIC spin column (Zymo Research C1011) for purification and DNase I treatment using the Zymo Direct-Zol system (Zymo Research R2050, R2060) following the manufacturer's guidelines. RNA quality was examined using spectrophotometry (NanoDrop 2000c, Thermo Scientific) and capillary electrophoresis (2100 Bioanalyzer, Agilent), and concentration was measured via fluorometric quantitation (Qubit HS RNA assay, Invitrogen Q32852). The final RNA product was stored at −80C.

Total RNA was reverse transcribed using the SuperScript IV first strand synthesis system (Thermo Fisher 18091050). 100 ng total RNA for each sample (n = 3 per time point) was converted to cDNA using oligo(dT)$_{20}$ primers. SYBR green assays (Applied Biosystems 4367659) were run in technical triplicate with 1 ng of cDNA template in each 12.5 µl reaction. Samples were amplified for 40 cycles using the LightCycler 480 II real time PCR system (Roche Diagnostics). All targets were normalized to *Gapdh* (see *Supplementary file 2* for primer sequences). Relative expression values were calculated for visualization using the $\Delta C_T$ method (*Livak and Schmittgen, 2001*); statistics were performed on $\Delta C_T$ values (*Supplementary file 1*). All self-designed primers were designed using PrimerBLAST (*Ye et al., 2012*).

## Statistics

For the RT-qPCR assays, statistical differences among the six timepoints were investigated via ANOVA and *post hoc* pairwise comparisons were computed using Tukey's Honestly Significant Difference test on the $\Delta C_T$ values (n = 3 biological replicates per time point; alpha = 0.05). R statistical software (*R Development Core Team, 2018*) was used for all RT-qPCR calculations and visualizations. Data analysis in R was facilitated using R packages included in the Tidyverse collection (*Wickham, 2017*) and statistical analysis was performed using the implementations of ANOVA and Tukey's HSD in 'stats' version 3.5.1 (*R Development Core Team, 2018*) . For each stage analyzed by flow cytometry, least three mice were used per group. Statistical differences between time points for all flow cytometry analysis were calculated using a Welch's t-test. One-way ANOVA was used to calculate statistical differences in cell density. A two-tailed t-test was used to calculate significance between BrdU incorporation in outer and inner tendon regions and Achilles tendon length measurements between different stages.

## Acknowledgements

We would like to thank Dr. Vladimir Vinarsky for help with 2-photon microscopy. We would also like to thank the MGH Center for Comparative Medicine for their services. JLG, HLD, KZ, MG, and TDC were supported by AR071554 NIAMS/NIH. JLG was supported by the American Federation of Aging Research, AR072294 NIAMS/NIH and the Harvard Stem Cell Institute. MG was supported by Human Frontier Science Program Fellowship. TDC was supported by the Milton Fund and Dean's Competitive Fund (Harvard University). HLD was supported by the NSF Graduate Research Fellowship Program. There was no additional external funding received for this study.

## Additional information

### Funding

| Funder | Grant reference number | Author |
| --- | --- | --- |
| National Institute of Arthritis and Musculoskeletal and Skin Diseases | AR071554 | Mor Grinstein<br>Heather L Dingwall<br>Ken Zou<br>Terence Dante Capellini<br>Jenna Lauren Galloway |
| National Institute of Arthritis and Musculoskeletal and Skin Diseases | AR072294 | Jenna Lauren Galloway |
| American Federation for Aging Research | | Jenna Lauren Galloway |
| Harvard Stem Cell Institute | | Jenna Lauren Galloway |
| Human Frontier Science Program | Fellowship | Mor Grinstein |
| Milton Fund | | Terence Dante Capellini |
| Harvard University | Dean's Competitive Fund | Terence Dante Capellini |
| National Science Foundation | Predoctoral fellowship | Heather L Dingwall |

The funders had no role in study design, data collection and interpretation, or the decision to submit the work for publication.

## Author contributions
Mor Grinstein, Conceptualization, Data curation, Formal analysis, Funding acquisition, Investigation, Visualization, Methodology, Writing—original draft, Writing—review and editing; Heather L Dingwall, Data curation, Formal analysis, Investigation, Methodology, Writing—original draft, Writing—review and editing; Luke D O'Connor, Investigation, Methodology, Writing—review and editing; Ken Zou, Data curation, Formal analysis, Investigation; Terence Dante Capellini, Formal analysis, Supervision, Writing—review and editing; Jenna Lauren Galloway, Conceptualization, Formal analysis, Supervision, Funding acquisition, Investigation, Writing—original draft, Project administration, Writing—review and editing

## Author ORCIDs
Mor Grinstein [ID] http://orcid.org/0000-0001-7166-5593
Heather L Dingwall [ID] https://orcid.org/0000-0003-2377-9777
Terence Dante Capellini [ID] http://orcid.org/0000-0003-3842-8478
Jenna Lauren Galloway [ID] https://orcid.org/0000-0003-3792-3290

## Ethics
Animal experimentation: This study was performed according to our protocol approved by the Massachusetts General Hospital Institutional Animal Care and Use Committee (IACUC: 2013N000062), and adheres to the recommendations in the Guide for the Care and Use of Laboratory Animals of the NIH.

## Decision letter and Author response
Decision letter https://doi.org/10.7554/eLife.48689.027
Author response https://doi.org/10.7554/eLife.48689.028

# Additional files

## Supplementary files
• Source code 1. RT-qPCR data analysis R script.
DOI: https://doi.org/10.7554/eLife.48689.021
• Source code 2. Rmarkdown file for generating RT-qPCR statistics tables.
DOI: https://doi.org/10.7554/eLife.48689.022
• Supplementary file 1. RT-qPCR Statistics tables for *Figure 4*.
DOI: https://doi.org/10.7554/eLife.48689.023
• Supplementary file 2. List of RT-qPCR primers used for *Figure 4*.
DOI: https://doi.org/10.7554/eLife.48689.024
• Transparent reporting form
DOI: https://doi.org/10.7554/eLife.48689.025

## Data availability
All data generated or analyzed in this study are included in the manuscript and supporting files. Source data and R code have been provided for Figure 4.

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
