## [Decision Letter]

Thank you for submitting your article "A distinct transition from cell growth to physiological homeostasis in the tendon" for consideration by *eLife*. Your article has been reviewed by two peer reviewers, and the evaluation has been overseen by a Reviewing Editor and Didier Stainier as the Senior Editor. The following individuals involved in review of your submission have agreed to reveal their identity: Hani Awad (Reviewer #1) and Christopher L Mendias (Reviewer #2).

The reviewers have discussed the reviews with one another and the Reviewing Editor has drafted this decision to help you prepare a revised submission.

Overall, there was enthusiasm for this manuscript and the authors attempt to define the temporal history of tendon cell turnover and proliferation from birth to aging using innovative model systems (BRDU labeling and H2B-GFP). The issues to address in the revision are relatively focused and experimentally relate to the following: 1-the role of ECM in cell turnover using 2 photon microscopy and DAPI; 2- quantitation of GFP by qRT-PCR; In addition the authors should note the limitations of aging the mice only to 20 months of age.

Reviewer #1:

This paper examines the natural history of cellular proliferation during post-natal growth and aging in axial tendons in mice. The paper presents elegant and state of the art approaches to genetically (using the H2B-GFP pulse-chase technique) and chemically (using BrdU incorporation) label proliferating tendon cells in post-natal, adult, and aging mice. One of the strengths of this manuscripts is the clarity of its objective and the simplicity of the research question. As the authors correctly identify, the temporal history of tendon cell turnover and proliferation and tehir contribution to the axial growth of tendon from birth to aging is a critical gap in knowledge in the field. Thus, the work is significant and important since all information on that question has thus far been obtained through indirect approaches. The study reports that the early stage after birth is characterized by high proliferative activity as determined by flow cytometric detection of CD45-negative and CD31-negative cells. This proliferative activity slows down in adults and aging tendons as tissue growth continues through ECM deposition and expansion resulting in a decrease in cell density as the tendon grows axially. In general, the experiments are well-executed, and the manuscript is well-written with few flaws that could be pointed out.

Major Comments:

Tendon harbors different populations of cells. The genetic pulse-chase labeling does not have the resolution to distinguish differences in cellular turnover in different cells without secondary genetic reporters or antibody labels. The BrdU-labeling study quantified Scx-Ai9+/Edu+ cells. Why didn't you use a similar approach for the genetic H2B-GFP pulse-chase experiment? Are there other genetic markers (transcription factors) associated with, say tendon stem cells, tendon-resident macrophages, or epitenon? What about Mkx or ECM-reporters (e.g. Col1, Fmod,.…) as secondary reporters. Please discuss.

The use of terminology related to cell-cycle transition should be eliminated since the study didn't really attempt to quantify cell cycle stages in any of the assays used.

The focus on cell proliferation while presenting new information is somewhat intuitive. Given that the continued growth of the tendon is cell proliferation-independent, a more in-depth analysis of ECM contributions is warranted. The paper presents a limited scope gene expression data set. Please discuss this limitation.

Regarding the above question, it might be feasible and less biased to determine cell density in a tendon volume imaged on a multiphoton microscope with DAPI to label the cells and SHG to define the ECM volume in 3D. This is technically possible given the small size of the mouse's tendon. Please consider using this approach or comment on the limitations of 2D histology.

The numbers on the y-axis of Figure 5B appear to be erroneous. A mouse's AT can't possibly be 10 – 60 mm (1 – 6 cm) long! Please review the data and revise the figures.

Reviewer #2:

The manuscript by Grinstein and colleagues evaluated tendon cell proliferation rates through 20 months of age in mice. This study enhances our understanding of the fundamental biology of tenocytes throughout a substantial portion of the lifespan of a mouse. The pulse-chase Dox H2B-GFP model combined with BrdU labeling was a thoughtful approach to study the expansion of cells from the originally labeled cells. While I think the study is well conducted, I have some points I would like the authors to consider:

– It would be useful to quantify GFP expression by qPCR as an additional quantitative marker to go along with Ki67, and support the quantitative analysis of GFP fluorescence in tissue sections. The issue of GFP intensity is important, as while the authors describe how they calibrated fluorescence with flow cytometry, this was not reported for microscopy and the images appear to be taken with a non-confocal microscope.

– Did the authors note any regional differences in GFP label retention or BrdU staining?

– The authors use dot plots for some data, but not others Since the sample size was low for some experiments, it would be more informative to present the data as dot plots throughout the manuscript.

– While the authors evaluated mice through 20 months of age, a limiting factor in this manuscript is that the maximum lifespan of a B6 mouse is ~36 months, and many of the pathological, aging-associated changes in tendon morphology do not occur until 24-26 months of age. I think this should be addressed as a limitation in this otherwise insightful paper.

---

## [Author Response]

Reviewer #1:

[…] In general, the experiments are well-executed, and the manuscript is well-written with few flaws that could be pointed out.Major Comments:Tendon harbors different populations of cells. The genetic pulse-chase labeling does not have the resolution to distinguish differences in cellular turnover in different cells without secondary genetic reporters or antibody labels. The BrdU-labeling study quantified Scx-Ai9+/Edu+ cells. Why didn't you use a similar approach for the genetic H2B-GFP pulse-chase experiment? Are there other genetic markers (transcription factors) associated with, say tendon stem cells, tendon-resident macrophages, or epitenon? What about Mkx or ECM-reporters (e.g. Col1, Fmod,.…) as secondary reporters. Please discuss.

The reviewer asked about using *Scx-Ai9*+ to quantify EdU and BrdU^+^ cells, but negative sorting of CD45/CD31 for the H2B-GFP+ population. They also mention the possibility of using different markers of cells resident in the tendon (stem cells, macrophages, epitenon, Mkx, or ECM reporters). We agree that it would have been ideal to use Scx-Ai9 to isolate the H2B-GFP cells, but our decision to use negative sorting for blood and endothelial cells is based on two reasons: we wanted to use an unbiased method to enrich for tendon cells independently of Scx-lineage and the crosses to generate the animals required the presence of 4 independent transgenes, which would have significantly increased the number of mice needed for the work as well as the time to generate the appropriate genotypes and age them to 18-20 months. We also agree that analyzing specific subsets of cells in the tendon would have been very informative regarding potential differences in turnover. We think that these are important points and we have added a sentence pointing to the difference in cells analyzed and a section discussing different cell types to paragraph two of the Discussion section.

The use of terminology related to cell-cycle transition should be eliminated since the study didn't really attempt to quantify cell cycle stages in any of the assays used.

The authors commented that we should remove terminology related to cell cycle transition since we did not quantify cell cycle stages. We agree with the reviewer and have changed all cell cycle terminology to cell proliferation changes, cell turnover, mitotic activity, or transitions in cell division rates.

The focus on cell proliferation while presenting new information is somewhat intuitive. Given that the continued growth of the tendon is cell proliferation-independent, a more in-depth analysis of ECM contributions is warranted. The paper presents a limited scope gene expression data set. Please discuss this limitation.

The reviewer commented that the gene expression data is limited in scope in particular in reference to the analysis of ECM markers. We have included a discussion of this limitation in the Discussion section.

Regarding the above question, it might be feasible and less biased to determine cell density in a tendon volume imaged on a multiphoton microscope with DAPI to label the cells and SHG to define the ECM volume in 3D. This is technically possible given the small size of the mouse's tendon. Please consider using this approach or comment on the limitations of 2D histology.

The reviewer suggested that 2-photon imaging to define 3D ECM and cell changes over time may be a better method to determine cell density given the limitations of 2D histology. To address this point, we have acquired 3D images of DAPI stained nuclei and SHG of collagen as well as *Scx-GFP* to quantify cell density and observe cell morphology in postnatal mouse tendons at P0, P7, P14, and P28. We have added this data to Figure 5A-E and video files 1-5. We did change the unit measurements to cells per 50 micron x 50 micron area, but the results are similar to our previous results in the changes observed between stages (original Figure 5A is now new Figure 5E). We detect a significant decrease in the cell density across each stage (Figure 5E), consistent with our original quantification. For these measurements, we counted cells in three 50 micron x 50 micron squares in 7-9 optical transverse sections per Achilles tendon per mouse with a gap of 20 microns between sections and at least 3 mice were analyzed per stage. The 50 micron x 50 micron square was created using FIJI (Schindelin et al., 2012), and in each optical section the nucleus number was counted in at least 3 different locations. Using the “Bright Z” mode, we kept the same 2-photon settings between sections and stages. Based on this, we can observe an increase in SHG signal from P0 to P28 (Figure 5A’’’-D’’’), indicating an increase in collagen organization and density. We also observe alterations in cell morphology from rounded to elongated, consistent with previous studies (Kalson et al., 2015).

The numbers on the y-axis of Figure 5B appear to be erroneous. A mouse's AT can't possibly be 10 – 60 mm (1 – 6 cm) long! Please review the data and revise the figures.

The reviewer noticed that our y-axis labels on Figure 5B are incorrect and in mm. This is absolutely correct, and we thank the reviewer for noticing this error. We have corrected the figure (new Figure 5F) and the text.

Reviewer #2:

The manuscript by Grinstein and colleagues evaluated tendon cell proliferation rates through 20 months of age in mice. This study enhances our understanding of the fundamental biology of tenocytes throughout a substantial portion of the lifespan of a mouse. The pulse-chase Dox H2B-GFP model combined with BrdU labeling was a thoughtful approach to study the expansion of cells from the originally labeled cells. While I think the study is well conducted, I have some points I would like the authors to consider:– It would be useful to quantify GFP expression by qPCR as an additional quantitative marker to go along with Ki67, and support the quantitative analysis of GFP fluorescence in tissue sections. The issue of GFP intensity is important, as while the authors describe how they calibrated fluorescence with flow cytometry, this was not reported for microscopy and the images appear to be taken with a non-confocal microscope.

The reviewer commented that we should quantify GFP expression by qPCR as an additional quantitative marker to Ki67 qPCR and flow cytometry data as our microscopy images were not taken on a confocal microscope. We agree that in order to compare the microscopy images, they should be taken on a confocal microscope under identical imaging conditions. We have since imaged new samples of Achilles tendons at P0 and P680 using 2-photon microscopy and identical imaging conditions (Figure 1B-C’’ and Materials and methods, which includes our explanation for using 2-photon rather than confocal microscopy). In regard to quantifying GFP expression by qPCR, this result will not match the protein expression that is observed in the microscopy images due to the Doxycycline-dependent transcription of the H2B-GFP transgene. The differences in H2B-GFP expression observed by flow cytometry is solely based on the dilution of the H2B-GFP protein rather than the mRNA levels, which are controlled by Doxycycline. We apologize that our description of this transgenic system was unclear and have since revised this in the text. The H2B-GFP system is comprised of a tetracycline/doxycycline transactivator that is constitutively expressed at the Rosa locus (Rosa:rtTA), and tetracycline/doxycycline operator elements in front of H2B-GFP (TetOP:H2B-GFP) (Foudi et al., 2009). This results in H2B-GFP being transcriptionally dependent upon the presence of Doxycycline-rtTA. As we only provided Dox from E10.5 to birth, transcription of H2B-GFP should be active at these stages and significantly reduced upon removal of Dox. The dilution of H2B-GFP observed by flow cytometry and microscopy is based on protein dilution as H2B is amongst the most stable proteins (Brennand et al., 2007; Toyama et al., 2013). Based on the reviewer’s comment, we felt that providing qPCR for GFP would not provide the appropriate information that the reviewer requested, and we felt that the core issue was proper imaging of the H2B-GFP tendons to qualitatively demonstrate dilution of the protein. To address this point, we have taken new images using identical 2 Photon settings by using the “Bright Z” mode to adjust the laser power between sections and samples. We hope that this new data addresses the reviewer’s concerns.

– Did the authors note any regional differences in GFP label retention or BrdU staining?

The reviewer asked if we noted any regional differences in the retention of H2B-GFP or BrdU staining. We think this is an excellent question and we did quantify the location of BrdU based on outer or inner tendon region incorporation and observed most of the incorporation occurred in the outer tendon regions. This new data is included and replaces the previous Figure 3F. The methods describing how these counts were performed are in the Materials and methods section (“Tendon length and cell counting measurements”).

– The authors use dot plots for some data, but not others Since the sample size was low for some experiments, it would be more informative to present the data as dot plots throughout the manuscript.

The reviewer asked that we present all data as dot plots. We have now changed Figure 3F and Figure 4 to dot plots.

– While the authors evaluated mice through 20 months of age, a limiting factor in this manuscript is that the maximum lifespan of a B6 mouse is ~36 months, and many of the pathological, aging-associated changes in tendon morphology do not occur until 24-26 months of age. I think this should be addressed as a limitation in this otherwise insightful paper.

The reviewer asked that we address the limitation that our aged mice were only examined at 20 months of age rather than 24-26 months when aged tendon phenotypes occur. We have now addressed this limitation in the Discussion section (paragraph five).

References:

Brennand, K., Huangfu, D., and Melton, D. (2007). All β cells contribute equally to islet growth and maintenance. PLoS Biol *5*, e163.

Toyama, B.H., Savas, J.N., Park, S.K., Harris, M.S., Ingolia, N.T., Yates, J.R., 3rd, and Hetzer, M.W. (2013). Identification of long-lived proteins reveals exceptional stability of essential cellular structures. Cell *154*, 971-982.